# Design and Control of a Tendon-Driven Robotic Finger Based on Grasping Task Analysis

**DOI:** 10.3390/biomimetics9060370

**Published:** 2024-06-19

**Authors:** Xuanyi Zhou, Hao Fu, Baoqing Shentu, Weidong Wang, Shibo Cai, Guanjun Bao

**Affiliations:** 1State Key Laboratory of Chemical Engineering, College of Chemical and Biological Engineering, Zhejiang University, 38 Zheda Road, Hangzhou 310027, China; zhouxuanyi@zjut.edu.cn (X.Z.);; 2Key Laboratory of Special Purpose Equipment and Advanced Processing Technology, Ministry of Education and Zhejiang Province, Zhejiang University of Technology, Hangzhou 310023, China2112002306@zjut.edu.cn (W.W.);; 3Zhejiang Ouyeda Machinery Manufacturing Co., Ltd., No. 333 Zhengdao Road, Huzhou 313000, China; 4College of Mechanical Engineering, Zhejiang University of Technology, Hangzhou 310023, China

**Keywords:** prosthetic hand, grasp, manipulation, human-inspired

## Abstract

To analyze the structural characteristics of a human hand, data collection gloves were worn for typical grasping tasks. The hand manipulation characteristics, finger end pressure, and finger joint bending angle were obtained via an experiment based on the Feix grasping spectrum. Twelve types of tendon rope transmission paths were designed under the N + 1 type tendon drive mode, and the motion performance of these 12 types of paths applied to tendon-driven fingers was evaluated based on the evaluation metric. The experiment shows that the designed tendon path (d) has a good control effect on the fluctuations of tendon tension (within 0.25 N), the tendon path (e) has the best control effect on the joint angle of the tendon-driven finger, and the tendon path (l) has the best effect on reducing the friction between the tendon and the pulley. The obtained tendon-driven finger motion performance model based on 12 types of tendon paths is a good reference value for subsequent tendon-driven finger structure design and control strategies.

## 1. Introduction

Multi-fingered, dexterous hands represent advanced mechanical systems designed to replicate the intricate functions and dexterity of the human hand. These hands are typically equipped with multiple joints and fingers, providing a high degree of freedom (DoF) [1,2]. This extensive DoF allows for a wide range of versatile operations, including grasping, rotating, and manipulating various objects encountered in daily life [3].

Multi-fingered dexterous hands can be categorized based on the form of transmission, gear rack mechanisms [4], worm gears [5], flexible shafts [6], tendons [7], timing belts [8], linkages [1], special materials like shape memory alloys (SMAs) [9], and so on. These diverse designs and configurations enable multi-fingered dexterous hands to function effectively in different scenarios, meeting the requirements of specific operations. Tendon-driven mechanisms (TDMs) are popular choices for the design of dexterous hands, which typically use inelastic ropes with minimal creep to simulate human hand tendons, offering the advantages of remote transmission and compliant transmission. There are numerous typical tendon-driven dexterous hands, for instance, the Okada Hand [10], the Utah/MIT hand [11], the ACT Hand [12] (Carnegie Mellon University), the Shadow Hand [13] (Shadow Robot the Company), and the DLR-I Hand [14]. Biomimetic tendon-driven prosthetic hands are also designed to imitate human hand movement. Narumi proposed a biomimetic prosthetic hand with bones, ligaments, tendons, and multiple muscles based on the human musculoskeletal system [15]. Liu designed a biomimetic finger based on anatomical structures actuated by SMA with hard and soft modes [16]. Additionally, Francisco presented a neurobiological control for anthropomorphic robotic systems to generate human-like movement [17].

A central focus in the study of tendon-driven mechanisms (TDMs) is the analysis of forces and structural design. Optimizing the tendon/pulley system to ensure purely rolling contact can significantly reduce friction [18]. Tendon/pulley system oriented in space that has purely rolling contact could reduce friction significantly [19]. Tendons pass through a path over ball-bearing pulleys mounted at each joint axis could reduce friction in the transmission system by 20% while increasing the maximum tip output force by 33% [20]. Huajie Hong [21] summarized applied tendon-driven topological architectures and the corresponding theoretical studies. Most TDM systems are based on N or N + 1 configurations, where N represents the number of DoF, as proposed by Morecki [22]. Salisbury [23] proposed that the choice of N-type configuration is based on the load requirements for transmitting rotary motion, typically conveyed through tensioned tendon sheaths, and such configurations generally incorporate springs to prevent reduction in system contraction. In N + 1 TDM systems, each rotating joint is connected to two tendons: one responsible for extension and the other for flexion [24]. TDMs could be decomposed into SubTDMs based on the bias force analysis [25]. To achieve precise manipulation performance, a tendon guiding structure, composed of soft and flexible materials, is proposed by Yong-Jae Kimet [26].

## 2. Related Work

The intricate architecture and functionality of the human hand serve as a profound source of inspiration for the design of robotic, dexterous hands [27]. The human hand’s tendons are primarily responsible for transmitting forces from muscles to bones, facilitating intricate and versatile movements. Tendons in the hand are divided into extrinsic and intrinsic groups [28].

Extrinsic tendons originate from muscles in the forearm and travel through the wrist to the fingers. These include the flexor and extensor tendons, which control the bending and straightening of the fingers.Intrinsic tendons originate from muscles within the hand itself. These include the lumbricals and interossei muscles, which fine-tune finger movements and contribute to the hand’s dexterity by enabling precise adjustments in finger positioning.

The independent, coordinated action of extrinsic and intrinsic tendons in the human hand facilitates a wide range of motions, from powerful grips to delicate manipulations [29]. The flexibility of tendon pathways in the human hand allows for complex, multi-joint movements. Robotic hands can benefit from flexible tendon pathways that enable multi-degree-of-freedom (DoF) motions. Designing tendon paths that optimize the human hand’s configuration can allow robotic fingers to achieve more natural and versatile movements, improving the hand’s adaptability to various tasks [30].

However, current studies on tendon transmission mechanisms are mostly based on specific finger mechanical structures and tendon configurations, so there is an urgent need to establish a set of universally applicable characteristic models for tendon transmission structures. Based on the classification of human hand grasping movements summarized by Feix [31], we conducted typical grasping tasks to identify the characteristics of human hand movements and establish evaluation metrics for dexterous manipulation. Evaluation metrics were used to describe the performance of tendon-driven fingers corresponding to 12 different types of tendon paths. The performance characteristics of tendon-driven fingers are of significant reference value for the structural design and control strategy of tendon-driven fingers, as well as for the manipulation planning of dexterous hands [32]. The innovations in the paper are summarized as follows:(1)A data collection glove is designed to collect the typical grasping tasks. The hand manipulation characteristics, finger end pressure, and finger joint bending angle are obtained through experiments based on the Feix grasping spectrum.(2)A series of contact motion characteristic comparison experiments were conducted on a tendon-driven finger with 12 different tendon pathways. These experiments revealed distinct output performance characteristics for each of the tendon pathways. The findings provide valuable insights that can inform the design and control strategies for future tendon-driven hands.

## 3. Human Hand Manipulation Characteristics

### 3.1. Structural Characteristics of the Human Hand

As shown in Figure 1, a normal and healthy human hand consists of 27 bones, including 14 phalanges, 5 metacarpals, and 8 carpals. Notably, the thumb is composed of two phalanges and one metacarpal, while each of the rest of the fingers consists of three phalanges and one metacarpal. According to the distance from the finger joints to the metacarpals, we refer to the phalanx segments from distal to proximal as the distal phalanx, the middle phalanx, and the proximal phalanx, respectively [33]. The distal and middle phalanges, as well as the middle and proximal phalanges, constitute rotating pairs capable of rotation within a plane, thereby contributing one DoF in space for each pair. The proximal phalanx and the metacarpal form a spherical pair, offering two DoFs in space, allowing for rotation in two directions [34]. Consequently, the thumb has five DoFs (two DoFs at the wrist), while each of the other four fingers has four DoFs. Collectively, these components contribute to the 21 DoFs of the whole human hand, crucial for facilitating dexterous manipulation.

Skeletal muscles can be divided into tendons and muscle bellies [28]. The muscle belly is capable of contraction; the tendon, located between the muscle belly and the skeleton, cannot contract but transmits the force generated by the contraction of the muscle belly to drive the skeleton. The thumb, relative to the other four fingers, is relatively independent in both position and tendon structure. Therefore, the movement of the thumb is not influenced by the other four fingers [35]. Each of the remaining four fingers experiences coupling when an adjacent finger moves, due to conjunct intrinsic hand muscles, thereby lacking fully independent movement.

### 3.2. Experimental Design of Human Hand Manipulation Characteristics

We present a design of a data collection glove tailored to acquire information on knuckle surface pressures and joint bending angles during human hand grasping. As shown in Figure 2, the glove is primarily composed of 14 pressure sensors and 5 bend sensors, capable of capturing contact pressures and joint bending angles of each finger segment at a frequency of 50 Hz during object grasping by the human hand.

In this study, five right-handed participants with normal hand function were selected to wear the glove and perform six typical grasping tasks summarized in the Feix grasping taxonomy, as shown in Figure 3. Each task was repeated five times, with knuckle surface pressures and joint angle variations recorded throughout the grasping process. The average values across the five trials were computed as the experimental results.

### 3.3. Analysis of Finger End Pressure

To accurately record the changes in the 14 finger end pressure data points, we labeled the ends of each finger. The thumb, index finger, middle finger, ring finger, and little finger were labeled “Th”, “In”, “Mi”, “Ri”, and “Li” respectively. The distal end, which is farthest from the palm, was labeled as one, the middle end as two, and the proximal end, which is closest to the palm, as three. For the thumb, the distal and proximal ends were labeled as one and two respectively. Using this method, the distal end of the index finger was recorded as In1. All 14 finger ends were numbered in this manner, and the final experimental results are shown in Figure 4.

From the experimental results, it is evident that in all of the grasping tasks, the thumb, index finger, and middle fingers are more actively involved, providing most of the grasping pressure. When grasping larger objects, the ring finger’s contribution to the grasping pressure increases, while the little finger’s involvement remains minimal, contributing almost no grasping force. The average grasping pressure exerted by each finger end during the grasping of various objects is summarized in Table 1. From the experimental results, it is evident that in all grasping tasks, the thumb, index finger, and middle finger are more actively involved, providing most of the grasping pressure. When grasping larger objects, the ring finger’s contribution to the grasping pressure increases, while the little finger’s involvement remains minimal, contributing almost no grasping force. The average grasping pressure exerted by each finger end during the grasping of various objects is summarized in Table 1.

According to the experimental results, the thumb, middle finger, and index finger are the primary contributors to the grasping force when grasping different objects, together providing approximately 80% of the total grasping force. This also confirms that a minimum of three fingers are required to achieve a stable grasp. Further analysis of the contact pressure distribution across the three ends of each finger reveals that the distal end bears a higher proportion of the contact pressure compared to the other two ends. Based on this finding, we established pressure distribution metrics for the human hand during grasping tasks.

### 3.4. Analysis of Finger Joint Bending Angle

The joint bending angle characteristics are recorded for grasping the cylinder with ϕ 45 mm, as shown in Figure 5.

By calculating the average value of each joint’s bending angle when the joint reaches its final stable state, the average maximum bending angle for each joint can be obtained, as shown in Table 2. Since the thumb has only two joints and plays a major role in almost all grasping tasks, the variation of the thumb’s joint angle is not considered.

The results indicate that when grasping a cylinder with ϕ 45 mm, the interphalangeal (IP) joint bending angles of each finger are generally greater than the MCP joints. Furthermore, by analyzing the slopes of the finger joint angle change curves in Figure 5, it is evident that the bending angular velocity of the IP joints is also greater than that of the MCP joints. During the grasping of a cylinder, the changes in joint angles typically follow a pattern of initial decrease followed by an increase, reflecting the natural action of finger extension prior to bending observed in actual grasping. This has significant implications for the control strategies of humanoid dexterous hands during object manipulation, as well as for the design of maximum bending angles for the finger joints.

## 4. Mechanical Analysis

### 4.1. Tendon Path Design

In tendon-driven finger systems, actuators are typically positioned in a base located away from the finger joints. The path that the tendons take from the base actuator to the driven finger joint directly affects the tension and speed of the tendons. In order to explore the relationship between different tendon pathways, different tendon pathways are designed to investigate the impact on the finger’s performance.

Each tendon pathway corresponds to a specific structural matrix [36]. The structure matrix of N + 1 type three-degree-of-freedom tendon-driven finger is defined as follows:(1)$$A_{k}=\left[\begin{array}{cccc} \pm R_{11} & \pm R_{21} & 0 & 0\\ \pm R_{12} & \pm R_{22} & \pm R_{32} & \;0\\ \pm R_{13} & \pm R_{23} & \pm R_{33} & \pm R_{43} \end{array}\right]$$
where $R_{ij}$ represents the radius of the pulley at the *j*-th joint that the *i*-th tendon wraps around. The sign preceding $R_{ij}$ is determined by the following rules:The direction of the axis of each finger joint is positive when it points outward from the plane of the hand.When a downward force is applied at the end of a tendon, if the pulley rotates counterclockwise around the joint axis, the force is positive; otherwise, it is negative.

In Equation (1), the number of rows represents the number of joints in the tendon-driven finger, while the number of columns represents the number of driving tendons. The structural characteristics must meet the following requirements:Each row must have at least two non-zero elements with opposite signs, ensuring that each joint of the finger can rotate in both directions.Swapping any two columns is equivalent to renaming the two tendons, which does not affect the overall functionality of the tendon-driven system.All zero elements in the structure matrix should be in the upper right corner of the matrix.Changing the sign of every element in any row does not affect the general characteristics of the matrix, which is equivalent to changing the positive direction of the joint axis.The rank of the structure matrix corresponds to the degrees of freedom of the system. For a system with *i*i tendons and *j*j degrees of freedom, at least one submatrix formed by deleting (*i* − *j*)(i − j) columns must have a non-zero determinant. If *i* = *j* + 1i = j + 1, then the determinant of the submatrix formed by deleting any column must be non-zero.If two structure matrices become identical after changing the sign of each element or rearranging some columns, they are considered structurally isomorphic. Structurally, isomorphic tendon pathways are regarded as the same.

For the N + 1 type three-degree-of-freedom tendon-driven finger, 12 tendon pathways are designed, as shown in Figure 6. Moreover, their structure matrices are calculated as follows:
(2)$$\begin{array}{cc} A_{a}=\left[\begin{array}{cccc} -R_{11} & R_{21} & 0 & 0\\ -R_{12} & -R_{22} & R_{32} & \;0\\ -R_{13} & R_{23} & -R_{33} & R_{43} \end{array}\right] & A_{b}=\left[\begin{array}{cccc} -R_{11} & R_{21} & 0 & 0\\ -R_{12} & -R_{22} & R_{32} & \;0\\ -R_{13} & R_{23} & R_{33} & R_{43} \end{array}\right]\\ A_{c}=\left[\begin{array}{cccc} -R_{11} & R_{21} & 0 & 0\\ -R_{12} & -R_{22} & R_{32} & \;0\\ -R_{13} & R_{23} & R_{33} & -R_{43} \end{array}\right] & A_{d}=\left[\begin{array}{cccc} -R_{11} & R_{21} & 0 & 0\\ -R_{12} & -R_{22} & -R_{32} & \;0\\ -R_{13} & R_{23} & R_{33} & -R_{43} \end{array}\right]\\ A_{e}=\left[\begin{array}{cccc} -R_{11} & R_{21} & 0 & 0\\ -R_{12} & -R_{22} & -R_{32} & \;0\\ -R_{13} & R_{23} & R_{33} & R_{43} \end{array}\right] & A_{f}=\left[\begin{array}{cccc} -R_{11} & R_{21} & 0 & 0\\ -R_{12} & -R_{22} & R_{32} & \;0\\ -R_{13} & R_{23} & -R_{33} & -R_{43} \end{array}\right]\\ A_{g}=\left[\begin{array}{cccc} -R_{11} & R_{21} & 0 & 0\\ -R_{12} & -R_{22} & R_{32} & \;0\\ -R_{13} & -R_{23} & -R_{33} & R_{43} \end{array}\right] & A_{h}=\left[\begin{array}{cccc} -R_{11} & R_{21} & 0 & 0\\ -R_{12} & -R_{22} & R_{32} & 0\\ -R_{13} & -R_{23} & -R_{33} & -R_{43} \end{array}\right]\\ A_{i}=\left[\begin{array}{cccc} -R_{11} & R_{21} & 0 & 0\\ -R_{12} & -R_{22} & -R_{32} & 0\\ -R_{13} & -R_{23} & R_{33} & R_{43} \end{array}\right] & A_{j}=\left[\begin{array}{cccc} -R_{11} & R_{21} & 0 & 0\\ -R_{12} & -R_{22} & -R_{32} & 0\\ -R_{13} & -R_{23} & R_{33} & -R_{43} \end{array}\right]\\ A_{k}=\left[\begin{array}{cccc} -R_{11} & R_{21} & 0 & 0\\ -R_{12} & -R_{22} & R_{32} & \;0\\ -R_{13} & -R_{23} & R_{33} & R_{43} \end{array}\right] & A_{l}=\left[\begin{array}{cccc} -R_{11} & R_{21} & 0 & 0\\ -R_{12} & -R_{22} & R_{32} & \;0\\ -R_{13} & -R_{23} & R_{33} & -R_{43} \end{array}\right] \end{array}$$

### 4.2. Structure Design of the Tendon-Driven Finger

The middle finger was selected as the reference for finger segment length proportions. The lengths of the distal, middle, and proximal phalanges of the tendon-driven finger were designed to be 20 mm, 30 mm, and 50 mm, respectively. Based on the range of motion of human finger joints, the rotation range of each joint on the tendon-driven finger experimental platform was set to [0, 90°].

Figure 7 shows the prototype of the tendon-driven finger. The entire tendon-driven finger system consists mainly of the finger section and the drive section. The finger section comprises three joints: the DIP joint, the PIP joint, and the MCP joint. Each joint is equipped with an angle sensor. The drive section consists mainly of the base, servo drive wheels, and servos. Each finger segment has perforations at the top, allowing tendons to pass through and be secured. The base is composed of four parts designed with mortise and tenon joints, ensuring tight connections and fixation. The base also features grooves that fit the finger base perfectly, securing the connection between the finger and the base.

### 4.3. The Tendon Tension Model

We adopt a stiffness–damping model to mechanically model the tendons. As shown in Figure 8, a typical tendon/pulley model is depicted, where pulley i is the driving pulley, and pulley j is the driven pulley. The tension $T_{i,j}$ in the tendon between pulley i and pulley j can be expressed as follows:(3)$$\begin{array}{l} T_{i,j}=T_{i,j}^{\prime }+k_{i,j}(r_{i}\theta_{k,i}-r_{j}\theta_{k,j})\\ +c_{i,j}(r_{i}\theta_{k,i}-r_{j}\theta_{k,j}) \end{array}$$
where $T_{i,j}^{\prime }$ is the initial tension of the tendon; $k_{i,j}$ is the stiffness of the tendon; $c_{i,j}$ is the damping of the tendon; $\theta_{k,i}$ is the wrap angle of the tendon around pulley i; and $\theta_{k,j}$ is the wrap angle of the tendon around pulley j.

### 4.4. Force Analysis of a Pulley

As shown in Figure 9, the pulley j is subjected to the tension denoted as $T_{j,j-1}$ and $T_{j,j+1}$, from two tendons, respectively. The vectors from the origin of the coordinate system j to the points tangent to the pulley and $T_{j,j-1}$, the pulley and $T_{j,j+1}$, are denoted as ${\vec{r}}_{j,j-1}$ and ${\vec{r}}_{j,j+1}$, respectively. The angles counterclockwise from $x_{i}$ and $x_{i+1}$ to the vector ${\vec{r}}_{j,j-1}$ and ${\vec{r}}_{j,j+1}$ are denoted as the wrap angle $\lambda_{j,j-1}$ of $T_{j,j-1}$ and $\lambda_{j,j+1}$ of $T_{j,j-1}$, respectively. Thus, the vectors can be represented as follows:(4)$$\begin{array}{c} {\;}^{i}{\vec{r}}_{j,j-1}={\left[r_{j}c\lambda_{j,j-1},r_{j}s\lambda_{j,j-1},b_{j,i}\right]}^{T}\\ {\;}^{i+1}{\vec{r}}_{j,j-1}={\left[r_{j}c\lambda_{j,j+1},r_{j}s\lambda_{j,j+1},b_{j,i+1}\right]}^{T} \end{array}$$
where $b_{j,i}$ is the distance along $z_{j}$ from the origin of the coordinate i to the origin of the coordinate j; $r_{j}$ is the radius of pulley j.

The unit direction vectors of tendon tension $T_{j,j-1}$ and $T_{j,j+1}$ are as follows, respectively:(5)$$\begin{array}{l} {\;}^{i}{\vec{u}}_{j,j-1}={\left[s\lambda_{j,j+1},-c\lambda_{j,j+1},0\right]}^{T}\\ {\;}^{i+1}{\vec{u}}_{j,j+1}={\left[-s\lambda_{j,j+1},c\lambda_{j,j+1},0\right]}^{T} \end{array}$$

During tendon-driven finger movements, it is possible for the axes of the pulleys at both ends of the linkage to not be parallel. As depicted in Figure 10, there exists a spatial torsion angle α between the axes of the two joints, necessary to introduce a rotation matrix $R_{x}$ around the *X*-axis:(6)$${\;}_{i+1}^{i}R_{x}={\left[\begin{array}{ccc} 1 & 0 & 0\\ 0 & c\alpha & -s\alpha \\ 0 & s\alpha & c\alpha \end{array}\right]}^{T}$$

The tension vector of the tendon in the coordinate system j can be represented as follows:(7)$$\begin{array}{l} {\;}^{j}{\vec{T}}_{j,j-1}=T_{i,j-1}{\;}^{j}R_{i}{\;}^{i}{\vec{u}}_{j,j-1}\\ {\;}^{j}{\vec{T}}_{j,j+1}=T_{i,j+1}{\;}^{j}R_{i+1}{\;}^{i+1}{\vec{u}}_{j,j+1} \end{array}$$

## 5. Experiments and Simulation of Tendon-Driven Fingers

In tendon-driven finger systems, the actuators are usually mounted on a base far away from the finger joints. To drive the distal joints of multi-DoFs fingers, the tendons must pass through multiple finger joints, and the path of the tendon from the base actuator to the driven joint directly affects the tension and speed of the tendon. As shown in Figure 11, for the N + 1 type tendon-driven finger with three DoFs, we designed 12 different tendon path configurations and explored the impact of the tendon path on the performance of the tendon transmission finger.

As shown in Figure 12, the stepwise motion of the three joints of the finger is realized using the IF motion function provided by Adams, enabling the three phalanges of the finger to contact the cylinder sequentially. Simulation is conducted to observe the fluctuations in tendon tension during the contact motion for the finger corresponding to the 12 different tendon path configurations.

Since the rotation of the DIP joint is the final motion action when a finger contacts an object, its driving tendon tension is generally unaffected by the rotation of other joints. Therefore, the fluctuations in DIP tendon tension are not considered in the simulation. Fluctuations in tendon tension significantly impact the grasping stability of the finger, with smaller fluctuations resulting in better stability. The magnitude of tension fluctuation can be described by comparing the standard deviation of each driving tendon’s tension, after it reaches its peak value, to the average tension. The experimental results are shown in Figure 13.

The results indicate that the tension fluctuation in the MCP driving tendon is generally greater than that in the PIP tendon. This discrepancy arises from both the DIP and PIP driving tendons passing through pulleys affixed to the MCP axis, resulting in a more pronounced coupling effect on the MCP joint. In the practical design of tendon-driven fingers, minimizing this scenario is advisable. Among the 12 tested tendon path configurations, tendon path (d) demonstrates the most ideal tension fluctuation, offering valuable insights for the development of tendon-driven fingers capable of stable grasping.

Evaluating the performance of tendon-driven finger movements across diverse tendon path configurations necessitates precise position control as a fundamental experiment shown in Figure 14. We adopt the N + 1 type tendon-driven scheme, where each joint is equipped with a dedicated driving motor. Joint rotation is achieved through tendon movement driven by winding pulleys fixed to the motor output shafts. While the motor rotates at a consistent angle, varying tendon path configurations yield different joint rotation angles. As illustrated in Figure 15, our experimental setup records joint rotation angles under various tendon path configurations. The motor output shaft rotates from 0° to 45° within 3 s, maintaining this position for 15 s, and then returns to 0° within another 3 s. These recorded angles serve as evaluation metrics for the motion performance of tendon-driven fingers. Changes in joint angles are captured by angle sensors positioned at each joint.

To ensure precise control of the joint position, the deviation between the joint rotation angle and the motor rotation angle is kept within 10%. Given the maximum motor rotation angle of 45°, the anticipated range for joint angles is calculated as [40°, 50°]. As depicted in Figure 15, the tendon paths between these two dotted lines conform to the acceptable error range.

The results show that only the tendon path (e) has all three joint rotation angles within the deviation range, with MCP, PIP, and DIP joint deviations of −4.4%, 8.9%, and 4.4%, respectively, demonstrating the effectiveness of using a tendon path (e) for joint position control. Tendon paths (d), (g), (j), and (l) achieve accurate control over two joints of the tendon-driven finger, which provides insights for designing tendon-driven fingers with high motion precision.

During transmission, the friction between the belt and pulley is primarily influenced by tension and the wrap angle of the pulley; a larger wrap angle typically results in increased friction. Leveraging existing knowledge of belt transmission, we investigate the factors influencing friction force in tendon transmission. Initially, experiments were conducted to establish the relationship between the wrap angle of the pulley and friction under varying tendon tensions. Subsequently, mathematical geometry methods were employed to calculate the wrap angles for each joint’s pulley under different tendon paths, facilitating the assessment of frictional force loss. This approach allows us to evaluate the force transmission efficiency of different tendon path configurations.

As shown in Figure 16, the experimental platform mainly includes a dynamometer, stepper motor, tendons, weights, pulleys with adjustable mounting positions, and aluminum profiles. The dynamometer is mounted on a lead screw platform. One end of the tendon is tied to the weight, and the other end is tied to the terminal of the dynamometer. The movement of the weights is achieved by the motion of the lead screw platform, and the difference between the value of the dynamometer during the uniform motion of the lead screw platform and the initial gravity of the weight represents the frictional force between the tendon and the pulley. By vertically adjusting the mounting position of the pulley and horizontally manipulating the aluminum profile, the wrap angle of the tendon around the pulley can be modified. Furthermore, altering the weight mass allows for adjustments in tendon tension.

As shown in Figure 17, the relationship between friction force and wrap angle in tendon/pulley system is depicted through a curve graph. Quadratic functions are used to fit the scatter plots of the tendon/pulley friction force against the wrap angle under three different tension conditions. The fitting functions are illustrated in the figure, where the values for the fitting curves under tension conditions of 200 g, 500 g, and 1000 g are 0.718, 0.961, and 0.869, respectively. The linearity between each data point and the fitting curve is relatively good. The results indicate that as the wrap angle of the tendon on the pulley increases, the frictional force between the tendon and the pulley also increases. Moreover, as the tension in the tendon increases, the rate at which the frictional force increases with the wrap angle accelerates. This provides valuable guidance for the subsequent selection of tendon paths in the design of tendon-driven fingers.

After calculating the wrap angles of the PIP and DIP driving tendons at each finger joint pulley and combining this with Figure 17, the relationship curves between friction force and wrap angle tendon/pulley were obtained. Moreover, frictional force losses in the PIP and DIP joint driving tendons under different tension conditions and different tendon paths were determined.

The results show that the minimum frictional force loss for the DIP joint driving tendons occurs in tendon path (a) under loads of 200 g, 500 g, and 1000 g, with respective force losses of 0.355 N, 1.056 N, and 1.928 N. For the PIP joint driving tendons, the minimum frictional force losses are found in tendon paths (c), (e), and (l), amounting to only 0.288 N, 0.587 N, and 1.727 N, respectively. These tendon paths can effectively reduce the frictional force between the tendon and pulley and achieve maximal joint output force if they are the actual choices. This has significant implications for the subsequent design of tendon-driven fingers with efficient force transmission.

## 6. Conclusions

This study contributes to advancing the dexterity of tendon-driven fingers by investigating the impact of tendon pathways on system performance. The main parts of this article can be summarized as follows:Drawing insights from human hand dexterity, we developed a data acquisition glove capable of capturing fingertip pressure and joint bending angles during Feix motion spectrum-based grasping tasks. This glove facilitated the collection of data on human hand characteristics when grasping different objects. From this analysis, we established human hand operational feature metrics, including fingertip pressure distribution and joint rotation angular velocity. These metrics provided a reference framework for evaluating the performance of tendon-driven fingers with 12 different tendon pathways.An experimental platform is designed for tendon-driven fingers to validate their performance. We evaluated various finger drive schemes and tendon transmission configurations, opting for the N + 1 drive and pulley-based tendon transmission. Validation experiments, as well as comparative motion characteristic experiments, are conducted for the 12 designed tendon pathways. The results revealed distinct performance characteristics for each pathway. Notably, pathway 4 exhibited the least fluctuation in tendon tension. Pathways 2, 11, and 12 met the contact pressure distribution requirements for the index finger, ring finger, and little finger, respectively. Pathway 1 satisfied the angular velocity metrics for finger joint rotation. Pathway 5 achieved the highest motion accuracy. Pathway 12 had the lowest friction losses.

Future research should expand the scope to encompass a wider range of daily hand operations, involving more participants to ensure comprehensive and accurate experimentation. Incorporating mathematical analysis and machine learning techniques holds promise for uncovering universal and valuable characteristics of human hand dexterity. Additionally, the integration of position and force closed-loop control strategies is anticipated to render movements more human-like.

## Figures and Tables

**Figure 1 biomimetics-09-00370-f001:**
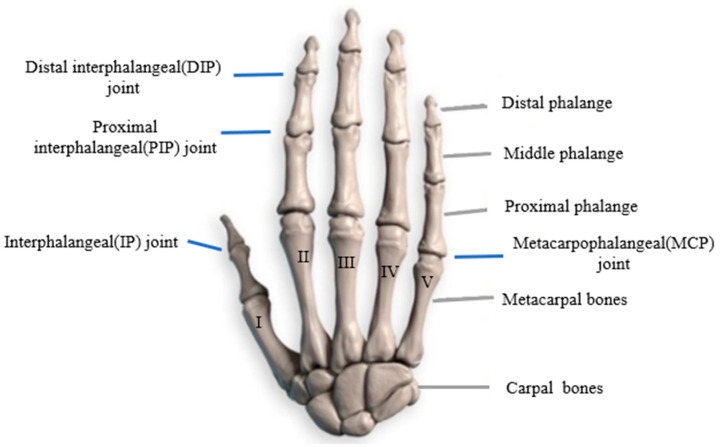
Skeleton structure of human hand.

**Figure 2 biomimetics-09-00370-f002:**
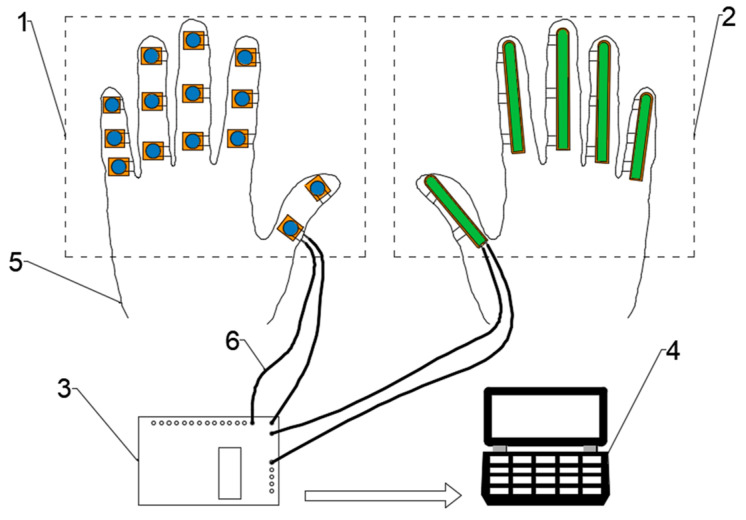
The schematic of the data collection glove. (1) finger pressure collection unit, (2) finger joint bending collection unit, (3) data acquisition board, (4) upper computer, (5) cotton gloves, (6) signal line.

**Figure 3 biomimetics-09-00370-f003:**
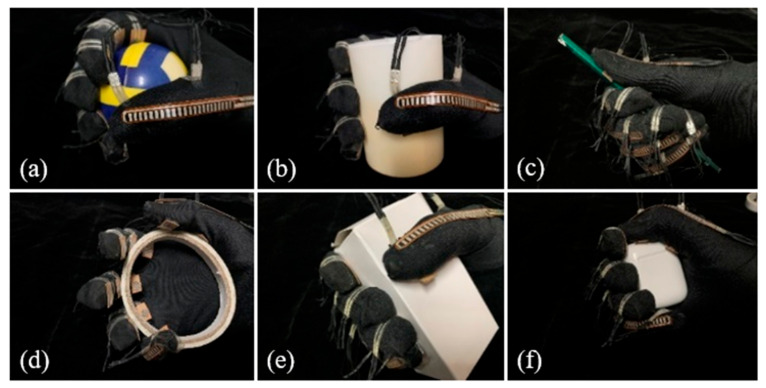
Grasping experiment. (**a**) A sphere with Sϕ 56 mm; (**b**) a cylinder with ϕ 45 mm; (**c**) a pencil with ϕ 7 mm; (**d**) a disc with ϕ 90 mm; (**e**) a rectangular prism with dimensions of 96 × 50 × 36 mm; and (**f**) a small rectangular prism.

**Figure 4 biomimetics-09-00370-f004:**
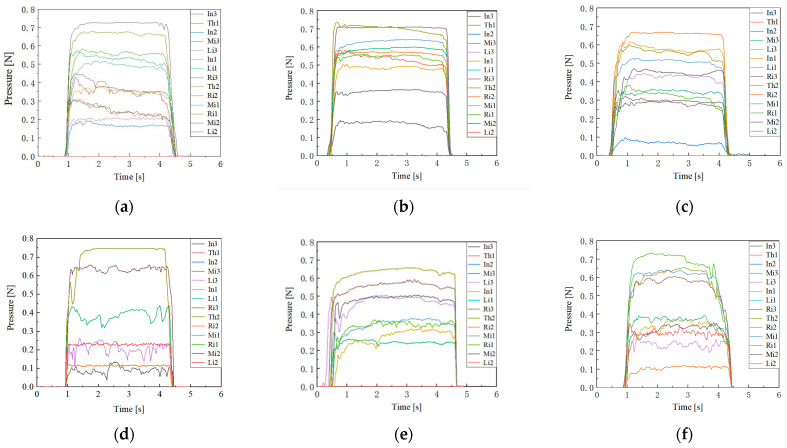
The pressures of 14 knuckle surfaces on distinct objects. (**a**) A sphere with Sϕ 56 mm; (**b**) a cylinder with ϕ 45 mm; (**c**) a pencil with ϕ 7 mm; (**d**) a disc with ϕ 90 mm; (**e**) a rectangular prism with dimensions of 96 × 50 × 36 mm; and (**f**) the small cuboid.

**Figure 5 biomimetics-09-00370-f005:**
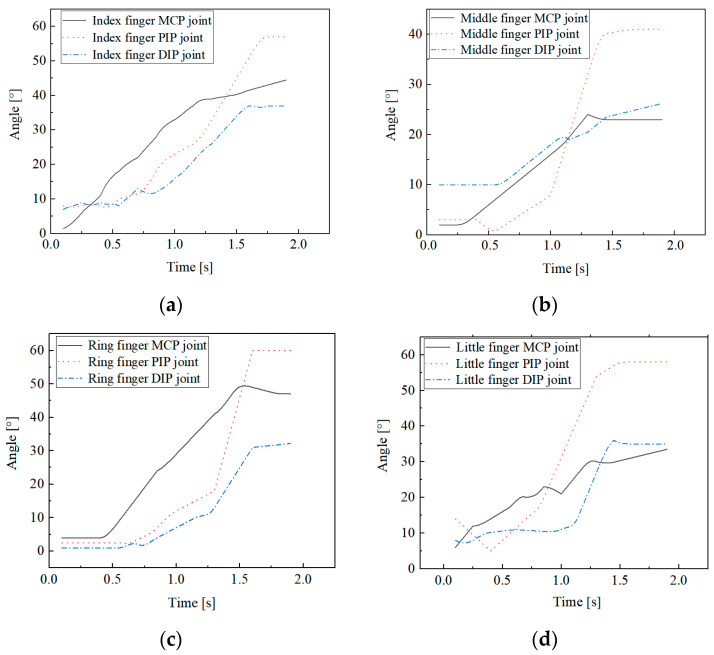
Finger joint angles for grasping a cylinder with ϕ 45 mm. (**a**) Index finger; (**b**) middle finger; (**c**) ring finger; and (**d**) little finger.

**Figure 6 biomimetics-09-00370-f006:**
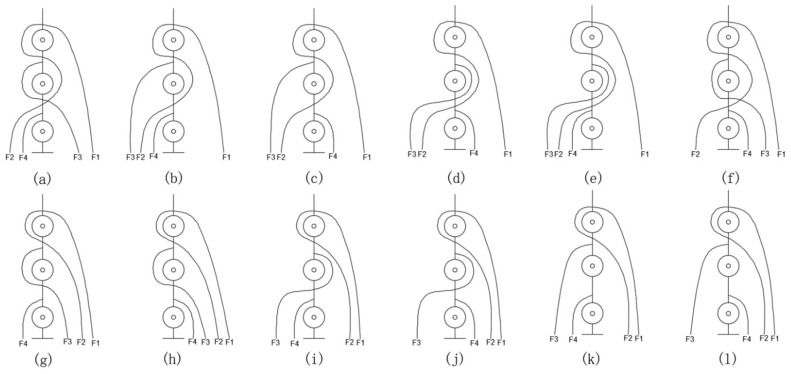
Schematic diagram of 12 tendon rope paths.

**Figure 7 biomimetics-09-00370-f007:**
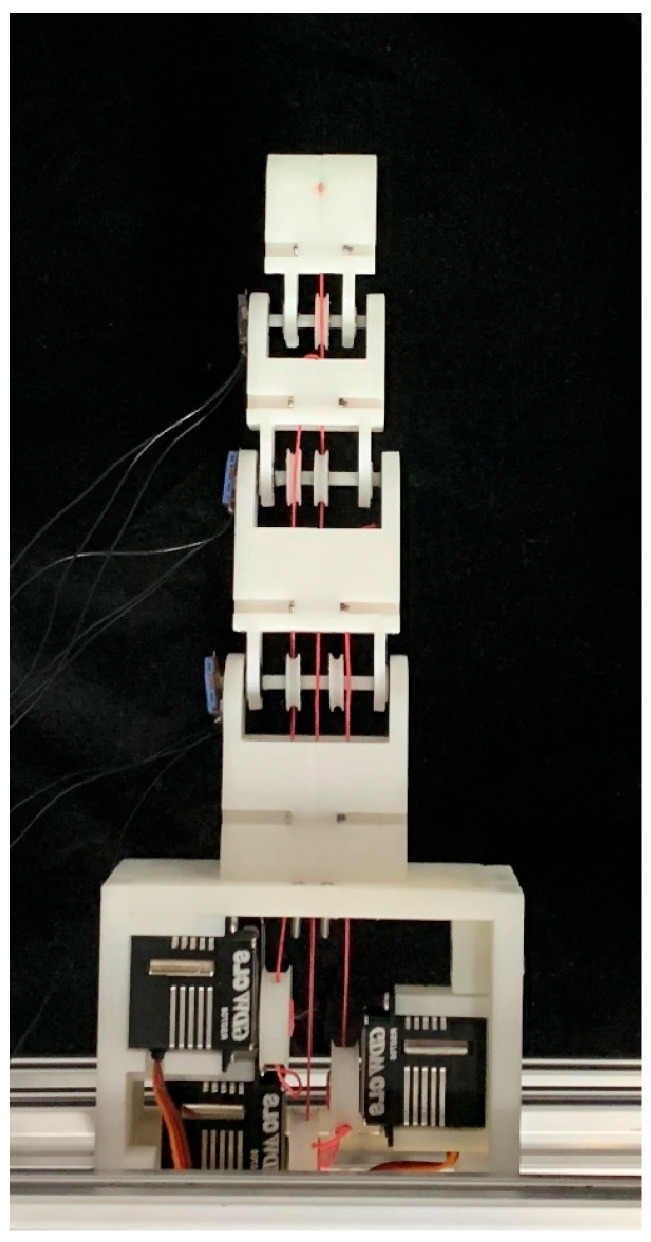
Prototype of tendon-driven finger.

**Figure 8 biomimetics-09-00370-f008:**
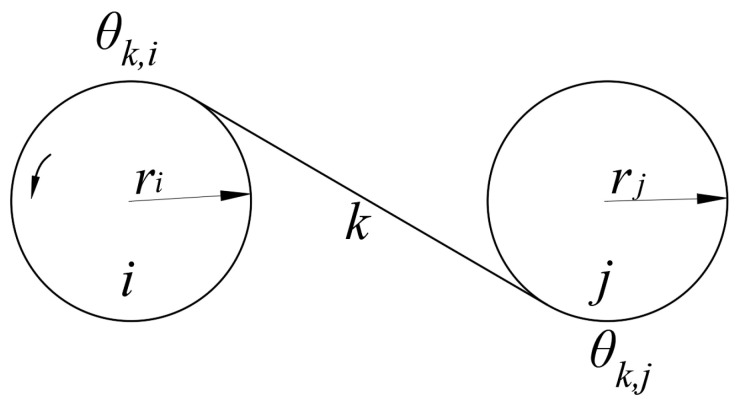
The tendon/pulley model.

**Figure 9 biomimetics-09-00370-f009:**
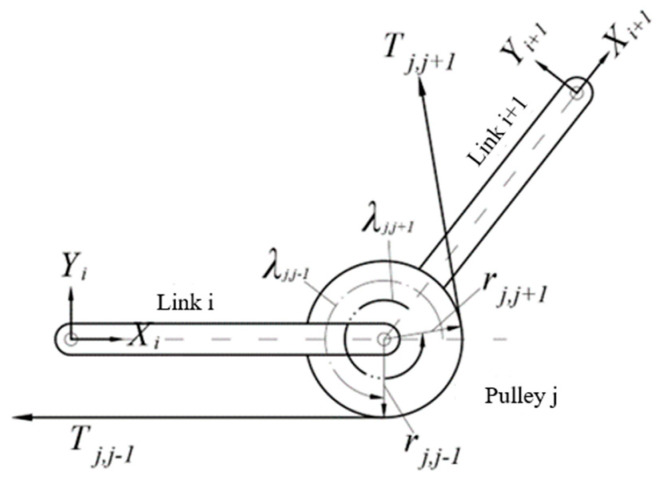
Force analysis of the pulley.

**Figure 10 biomimetics-09-00370-f010:**
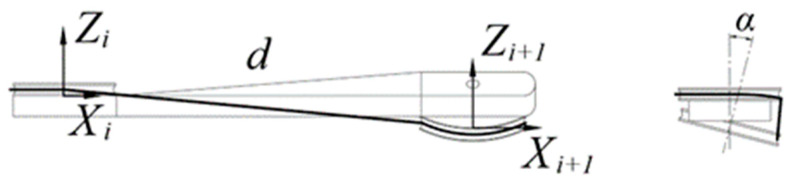
Linkage with twist angle.

**Figure 11 biomimetics-09-00370-f011:**
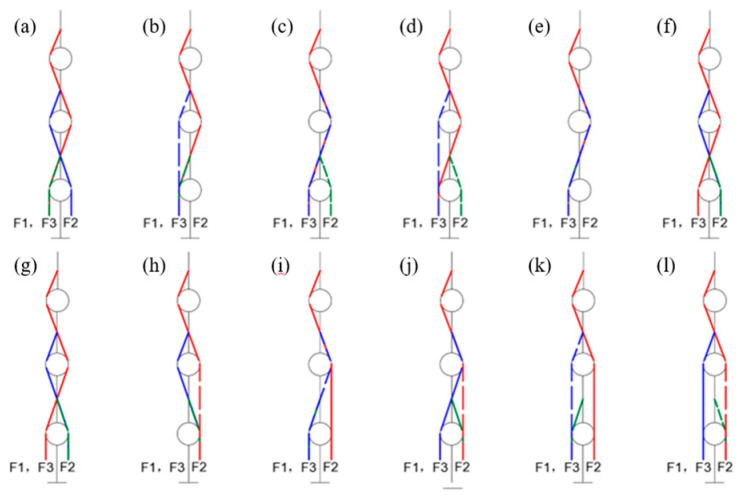
Schematic representation of 12 tendon paths numbered from (**a**–**l**).

**Figure 12 biomimetics-09-00370-f012:**
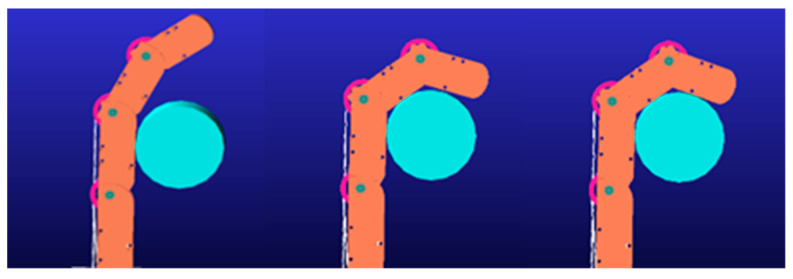
Simulation of the fluctuations in tendon tension.

**Figure 13 biomimetics-09-00370-f013:**
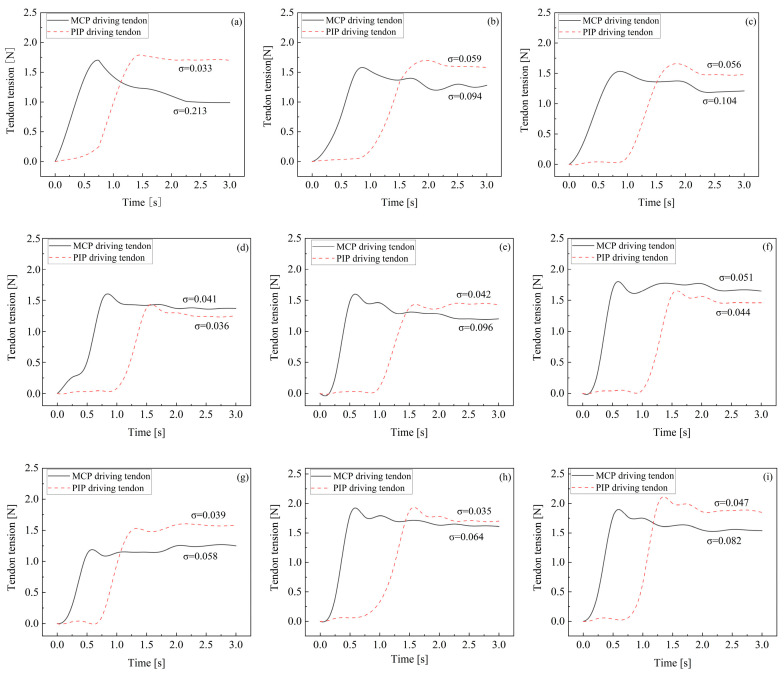
The fluctuations of tendon tension of 12 paths configurations numbered from (**a**–**l**).

**Figure 14 biomimetics-09-00370-f014:**
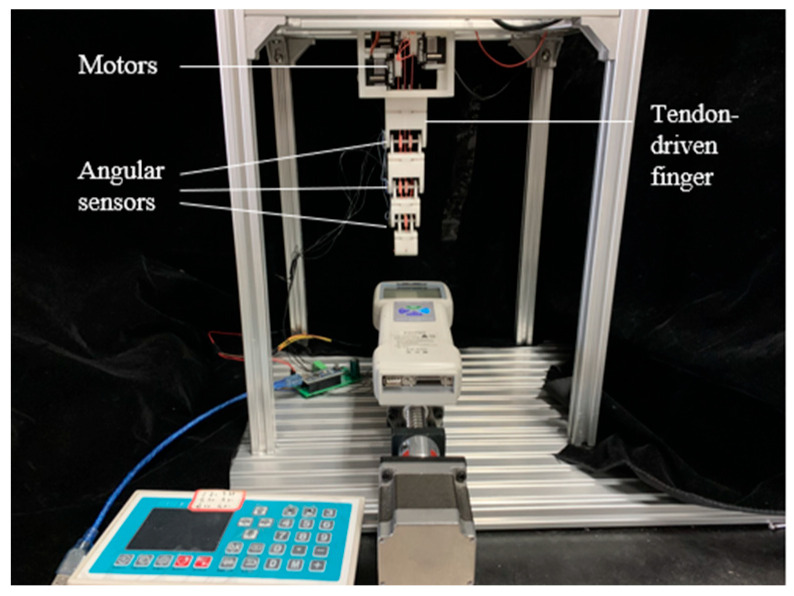
Experimental platform for the control of joint angle.

**Figure 15 biomimetics-09-00370-f015:**
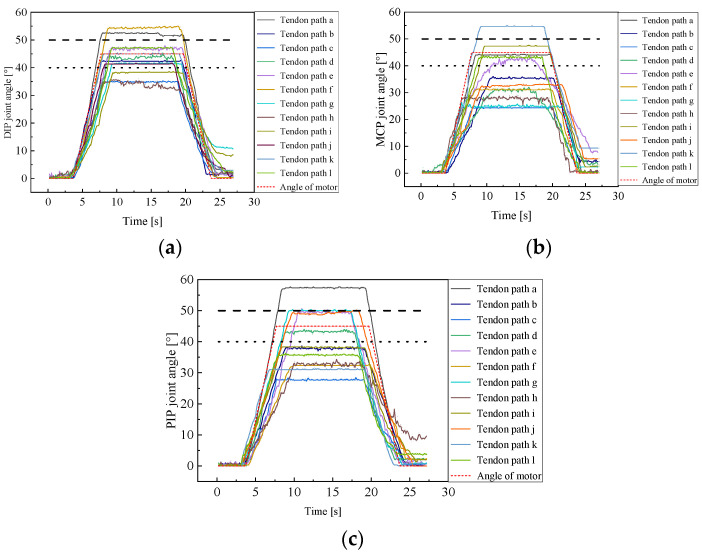
Position control experiment of finger joints. (**a**) DIP joint; (**b**) MCP joint; (**c**) PIP joint.

**Figure 16 biomimetics-09-00370-f016:**
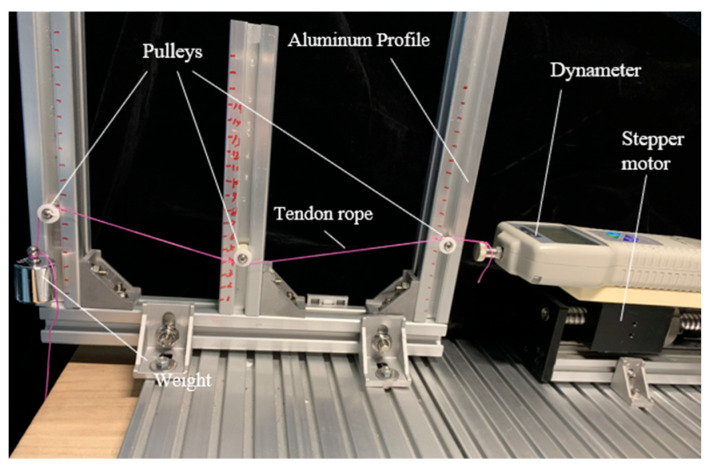
Experimental platform for verifying the relationship between friction force and wrap angle in tendon-pulley system.

**Figure 17 biomimetics-09-00370-f017:**
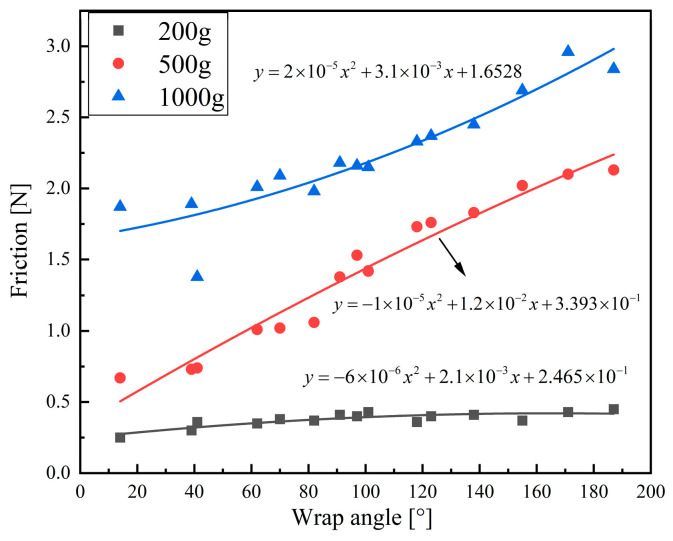
The relationship between friction force and wrap angle in tendon/pulley systems.

**Table 1 biomimetics-09-00370-t001:** The proportion of contact pressure on the end of each knuckle during human hand grasping.

Object	Thumb	Index Finger	Middle Finger	Ring Finger	Little Finger
Sphere with Sϕ 56 mm	31%	21%	26%	18%	4%
Cylinder with ϕ 45 mm	35%	24%	30%	15%	6%
Pencil with ϕ 7 mm	28%	25%	25%	12%	10%
Disc with ϕ 90 mm	41%	14%	15%	17%	13%
Rectangular prism 96 × 50 × 36 mm	39%	21%	24%	10%	6%
Small rectangular prism	21%	29%	33%	17%	0

**Table 2 biomimetics-09-00370-t002:** The maximum bending angle for each joint.

	Index Finger	Middle Finger	Ring Finger	Little Finger
Metacarpophalangeal (MCP) joint	39.97°	45.13°	49.85°	29.13°
Proximal interphalangeal (PIP) joint	62.60°	69.38°	70.40°	62.33°
Distal Interphalangeal (DIP) joint	46.58°	49.23°	35.97°	41.50°

## Data Availability

Data is contained within the article—The original contributions presented in the study are included in the article, further inquiries can be directed to the corresponding author.

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
