# Peer review of "Design and Control of a Tendon-Driven Robotic Finger Based on Grasping Task Analysis"

_biomimetics, 2024, doi:10.3390/biomimetics9060370_

Round 1

Reviewer 1 Report

Comments and Suggestions for Authors

This thesis develops a tendon-driven robotic finger for dexterous manipulation. Twelve tendon paths were designed and simulated, and the range of motion and force of each joint were measured by controlling the robot finger using the selected tendon path. 

This study can be good for providing a model for designing and evaluating N+1 type tendon transmission models.

However 

1. the relationship between the knowledge of human anatomy cited in the paper and the proposed model is weak. Rather, there are previous studies that used more anatomical knowledge in the following papers.

Narumi et.al. (2022); https://doi.org/10.3390/act11060167

Matsuoka et.al.(2006); Neurosurgical focus 20.5 1-9.

2. the size of the robot finger model and the diameter of the pulley used in the simulation are not described, and although it can be a way to evaluate which model is good for different finger lengths and diameter, the proposed path (d) is only good for a specific length, not a general one.

It is also necessary to consider whether the chosen path(d) is anatomically similar or different from the human.

3. Make sure to check if the references are cited correctly in line with journal guidelines.

4. the figure captions lack explanation: the abbreviations in the figures are not explained.

5. 'manipulability ellipsoid' is not explained.

Comments on the Quality of English Language

It seems that moderate editing of the English language is required.

Author Response

Dear Biomimetics Editorial Office、

Thank you very much for your letter informing us of the editorial decision on our manuscript (biomimetics-2994377) entitled " Motion Analysis of Tendon-driven Finger Based on Human Hand Manipulation Characteristics". The paper has been duly revised according to the comments made by the Editor and the Reviewers. We would like to resubmit the revised version for consideration of possible publication as a regular paper. The authors would like to express their sincere appreciation to the Editor for her/his constructive comments and the effort and the time spent helping us to improve the quality of the paper. In the following, we provide a specific response to the comment, explaining how the paper is revised. We have highlighted all the changes in red color in our revised manuscript. We hope that the revised manuscript addressed the reviewers' concerns in a satisfactory way.

Sincerely,

Authors

Reviewer 2 Report

Comments and Suggestions for Authors

The paper introduces a tendon-driven robotic finger, and presents a dexterous finger model and its control strategies by referring to the motion analysis obtained by experiments of typical grasping tasks using a data collection glove. The reviewer finds the research and the methods are interesting, however has the following questions and comments.

1) Does the title of the paper properly express the contents of the study? The paper seems to describe a tendon-driven robotic finger and its control strategy by referring to the analysis of typical grasping tasks using a data collection glove. The reviewer suggests the authors to modify the title to clearly present the purpose/contents of the study.

2) In Chapter 2, the structural characteristics of a human hand is introduced, however it is difficult to sufficiently understand the details of the mechanism to realize the dexterity of fingers. The authors are suggested to add relevant references including medical and anatomical papers.

3) In Line 75, the authors mention that “Skeletal muscles can be divided into tendons and muscle bellies”. As far as the reviewer recognizes, a skeletal muscle attaches to the bone by tendons, and they produce all-body movements. The authors are able to refer to the information from “Anatomy, Skeletal Muscle” at
https://www.ncbi.nlm.nih.gov/books/NBK537236/

4) From the reviewer’s experience, it is usually important to consider the effect of the side of tendons for understanding the dexterous finger motion. In fig.10, the reviewer is curious about the palmar and dorsal side of the robotic finger.

5) Twelve different combinations of tendon paths are discussed, however, the configuration in a normal human finger is not clearly described to be associated. The authors are suggested to add further discussions by referring to a normal human finger, and to emphasize the differences between the optimal schema (e) and a normal finger.
The authors may refer to the following references
https://www.mdpi.com/1424-8220/24/9/2924
https://www.mdpi.com/2313-7673/9/3/151

6) The experiment discussed in Fig.15 seems to have a longer distance than a human/robotic finger, and the authors are suggested to discuss this distance. If this affects the result, the analysis of friction should refine the model that has closer dimensions to a normal finger.

7) All through the manuscript, the authors should carefully check the figures and the corresponding descriptions in the texts. There are many improper descriptions such as;

- In line 97-98, the description of Fig.3 seems to be strange. What is a “Sϕ56mm”?
- Words “phalanx” and “phalanges” are both used. They should be consistent.
- There are some formatting errors at the start of page 5, around line 120. Maybe it is the missing Table 2?
- In line 235, the word “N+1” appears without the definition.
- In Fig.11, the difference between the 2nd and the 3rd figures cannot be distinguished due to the low resolution of provided PDF file. Please emphasize the differences by adding indicators.
- Most graphics, including Fig.5, 12, 14, have low resolution. The authors are suggested to used higher resolution and contrast for readability. Also, try to magnify the images for better readability.
- The title of Fig. 12 should clearly indicate that the results are simulated.
- In Line 266, what is “… , e wherein each …”? Please refine the sentence.
- Fig. 13 is not referred to in the text.
- Fig.15 seems to be different from the title, please carefully check everything again.
- In line 234-237, maybe the authors mean Fig.10 rather than 11. Please also check if there are any missing figures, tables or paragraphs.

Comments on the Quality of English Language

The authors are suggested to check grammatical errors carefully.

Author Response

(The authors gave the same response as above.)

Reviewer 3 Report

Comments and Suggestions for Authors

This manuscript presents the “ Motion Analysis of Tendon-driven Finger Based on Human Hand Manipulation Characteristics and this topic of the manuscript is interesting. The study provides valuable insights into understanding the structural basis of human hand dexterity and designing tendon-driven fingers. By analyzing the structural characteristics of human hand bones, joints, and tendons, the research establishes a foundation for enhancing hand dexterity. The development of evaluation metrics for human hand dexterity, based on typical grasping tasks and the Feix grasping spectrum, offers a standardized approach for assessing hand functionality. The exploration of twelve tendon rope transmission paths under the N+1 type tendon drive mode sheds light on the crucial role of tendon path design in hand motion performance. The experiment highlights distinct advantages among the different tendon paths, such as improved control over tendon rope tension, joint angle, and reduced friction between the tendon and pulley.

 Please check as following comments and hope helpful for improvement of the manuscript. These comments can assist readers in understanding main contributions quickly.

 1.It's recommended to provide more specific descriptions of the methods used in the abstract or content to enhance readers' understanding of the research approach.

 2.The research results can be detailed in the abstract with more sentences and also provide more quantitative data to assist readers in understanding main contributions quickly.

 3.Please state the significant impact of tendon path design on hand motion performance, further exploration and optimization of tendon path configurations are warranted. Experimentation with variations in path curvature, material properties, and attachment points could lead to enhanced control and efficiency.

 4.Please state sensors how to operate with the control system for tendon-driven finger designs for more details in this study?

Author Response

(The authors gave the same response as above.)

Round 2

Reviewer 1 Report

Comments and Suggestions for Authors

Many parts of the paper have been enhanced in content, but the overall structure of the text remains difficult to read.

1. For instance, abbreviations or information related to figures should be placed in the figure captions (or legends) rather than in the main text to improve readability.

2. The section numbers are incorrect. Introduction must involve '2. related works'. The section 3 is missing.

3. Additionally, the lack of a dedicated discussion section and the scattered placement of discussion points within the method and results section reduce the paper's readability.

4.The references cited in the text are currently mixed in different formats. Please ensure that the citation style is unified according to the journal submission guidelines.

Comments on the Quality of English Language

The overall document structure needs to be rearranged.

Author Response

Dear Biomimetics Editorial Office、

Thank you very much for your letter informing us of the editorial decision on our manuscript (biomimetics-2994377) entitled " Design and Control of a Tendon-Driven Robotic Finger Based on Grasping Task Analysis". The paper has been duly revised according to the comments made by the Editor and the Reviewers. We would like to resubmit the revised version for consideration of possible publication as a regular paper. The authors would like to express their sincere appreciation to the Editor for her/his constructive comments and the effort and the time spent helping us to improve the quality of the paper. In the following, we provide a specific response to the comment, explaining how the paper is revised. We have highlighted all the changes in red color in our revised manuscript. We hope that the revised manuscript addressed the reviewers' concerns in a satisfactory way.
